# INSIGHTS INTO THE MECHANISM BEHIND REUSING TEACHER'S CLASSIFIER IN KNOWLEDGE DISTILLATION

**Kinshuk Dua**
Independent Researcher
kinshukdua@gmail.com

## ABSTRACT

Knowledge distillation (KD) has emerged as an effective approach to compress deep neural networks by transferring knowledge from a powerful yet cumbersome teacher model to a lightweight student model. Recent research has suggested that re-using the teacher's final layer (i.e., the classifier) can be a straightforward and effective method for knowledge distillation. The underlying mechanism for this method's success remains unclear. Our study aims to shed light on how the knowledge distillation loss affects the alignment between the weights of the student classifier and the teacher classifier. Specifically, we compare the $L^2$ norm of the difference between the weights of the student and the teacher classifier during the training process. Our experiments demonstrate that the knowledge distillation loss might encourage alignment between the student and teacher classifiers, as indicated by a strong positive correlation ($> 0.97$) between the $L^2$ norm and the loss during training, although the size of this effect is small. We also observe that as temperature increases, this alignment seems to decrease and the $L^2$ norm behaves similar to normal (non-KD) training. These preliminary insights aims to provide to a better understanding of knowledge distillation provide a starting point for the development of new KD frameworks.

## 1 INTRODUCTION AND MOTIVATION

Knowledge Distillation aims to enhance the generalization capability of a less-parameterized student model by transferring the knowledge from a larger, more powerful teacher model. The straightforward method to achieve this is by aligning the logits of the student to the teacher using Kullback–Leibler divergence. (Hinton et al., 2015). However, there still remains a significant gap between the capabilities of the student and the teacher. In recent years, there have been numerous works around this problem that aims to eliminate this gap between the student and the teacher. These include complex knowledge distillation frameworks using intermediate features as addition supervision (Romero et al., 2015; Ahn et al., 2019; Chen et al., 2021), transferring attention (Komodakis & Zagoruyko, 2017), maximizing mutual information (Ahn et al., 2019), and using contrastive representation (Tian et al., 2020). Recent works has demonstrated that re-using the final layer provides a promising alternative to more complex knowledge distillation frameworks. This approach has been applied not only to classification tasks (Yang et al., 2021; Chen et al., 2022), but also to other computer vision tasks such as face recognition (Shi et al., 2020). However, research on this idea is still limited.

Yang et al. (2021) and Chen et al. (2022) both match the the penultimate representation of the student to the teacher using the $L^2$ loss to reuse the teacher's classifier. This helps increase because the predicted class probability is proportional to the alignment of the pre-final layer's output with the weights of the teacher's classifier (Nayak et al., 2019). However, The fundamental mechanism behind this approach is yet to be fully understood and has not been explored in literature. We wanted to look at vanilla knowledge distillation from a different perspective and ask discuss a few important questions, to serve as the motivation behind the recent works with a self contained experiment.

## 2 EXPERIMENTAL SETUP AND DISCUSSION

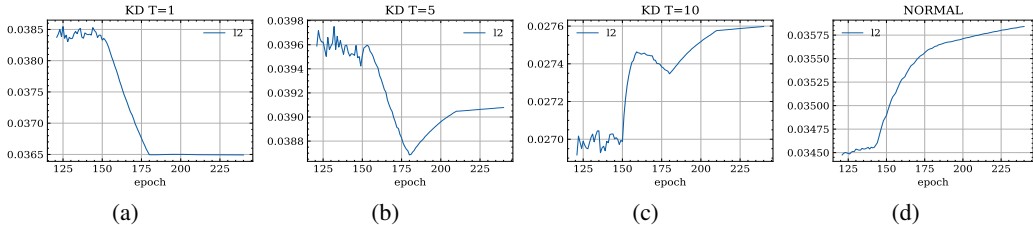

(a)          (b)          (c)          (d)

Figure 1: The $L^2$ norm of the differences between weights of the classifier the student and teacher during training for different Temperatures (T).

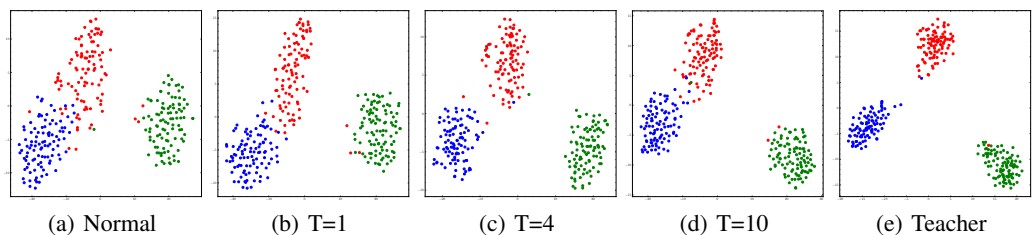

(a) Normal          (b) T=1          (c) T=4          (d) T=10          (e) Teacher

Figure 2: TSNE of the intermediate representations in the penultimate layer for (a) Normal Training (b) KD T = 1 (c) KD T = 4 (d) KD T = 10 (e) Teacher

We use the standard training setup as mentioned previous works (Chen et al., 2022; Hinton et al., 2015; Chen et al., 2021). We ran vanilla KD training on the CIFAR-100 dataset with temperature (T) ranging from 1 to 10 and a fixed $\alpha$ of 0.9. More details on the reproducibility are in the appendix. Let $\mu = ||w_t - w_s||$, where $w_t$ and $w_s$ are the weights of the teacher and student classifier. Let $\mathcal{L}$ be the combined cross-entropy and KL loss. We found that the $L^2$ norm is a simple and useful metric for comparing the misalignment between the weights. During the initial phase of the training, the learning rate is high and the weights increase rapidly, this is followed by huge fluctuations in the $\mu$, therefore we only report graphs after warmup epochs (120). Figure 1 shows the trend of $\mu$ for certain temperatures. Figure 2 showcases the visualization of the representations vectors *at similar accuracy (91 ± 0.5%)* from the penultimate layer using TSNE Van der Maaten & Hinton (2008). The graphs for all temperatures are in the appendix. We try to answer the following question based on the results. **1) Does the student and the teacher classifier align during vanilla Knowledge Distillation without an explicit representation matching loss?** To a degree yes, as seen in Fig. 1 (a), $\mu$ rapidly decreases and then settles down. We report a strong positive correlation (0.972, $p < 10^{-70}$) between $\mu$ and $\mathcal{L}$. This stands in contrast to normal training where there is a strong negative correlation (-0.971). **2) What about the effects of Temperature?** As temperature increases, the correlation between $\mu$ and $\mathcal{L}$ the seem to rise, peaking at around T=4, then rapidly decreasing until it becomes strongly negative similar to normal training. The graph of correlation with different temperature has also been included in the appendix for the sake of completeness. We believe this might be due to the fact that as $T \rightarrow \infty$, the KL Loss behaves similar to the MSE Loss (Kim et al., 2021). **3) Does Knowledge Distillation implicitly match the student's penultimate representation to the teacher?** We see that the representations of the students *at similar accuracy* to normal training is significantly better, we also note that the inter-class similarity seemed to increase with temperature. At T=10 (Fig. 2 (d)), the increased inter-class similarity seems to come at the cost of a decreased intra-class dissimilarity. The distance between different classes is highest at around T=4 (Fig. 2 (c)), which correlates why a temperature of 4 had the highest accuracy in reported works. We believe our analysis on this self-contained experiment, has been able to shed some light on the internal mechanics of knowledge distillation, especially in terms of the final layer of the student.

## URM STATEMENT

All the authors of this work meets the URM criteria of ICLR 2023 Tiny Papers Track.

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

## A APPENDIX

### A.1 FURTHER DISCUSSION

We selected the $L^2$ norm as it provided the most optimal results, while cosine similarity was initially promising and yielded similar results to the $L^2$ norm. Unfortunately, the output demonstrated considerable fluctuations and variability. Due to limited resources, we were unable to conduct experiments on ImageNet and on other model architectures like ViT and InceptionNet. We also referenced

a related study Huang et al. (2022), which exhibited similar behavior despite using a different loss, but this was beyond the scope of our paper. In addition, we performed preliminary ablation experiments on $\alpha$, the KL loss to cross-entropy loss ratio. However, changing the ratio did not significantly affect the overall correlation and thus the results of our paper. Although the ratio did lower the magnitude of $L^2$ norm $\mu$ (indicating increased alignment as anticipated), it did not appear to alter the general shape of $\mu$ over the epochs.

You can find the animated plot of the $L^2$ norm for different temperatures in this repository along with other supplementary material.

## A.2 LIMITATIONS

- There is the possibility that the student and teacher might have learned equivalent classifier layers, but that they would still not perfectly align (i.e. the $L^2$ plateaus instead of approaching 0) because one is a permutation of the other. Further analysis is needed for exploring the same.

- It is worth keeping in mind that that t-SNE can give misleading answers in some cases, especially when evaluating distances between clusters.

- We decided to run the tests without weight regularization to reduce the number of confounding factors, due to which the size of the weights increased rapidly especially in the first few epochs making the effects difficult to visualize. Therefore we decided to dropped results before epoch 120 (warmup epochs) in the graphs. This however, may lead to an overestimate of the size of the effect.

## A.3 EXTRA GRAPHS

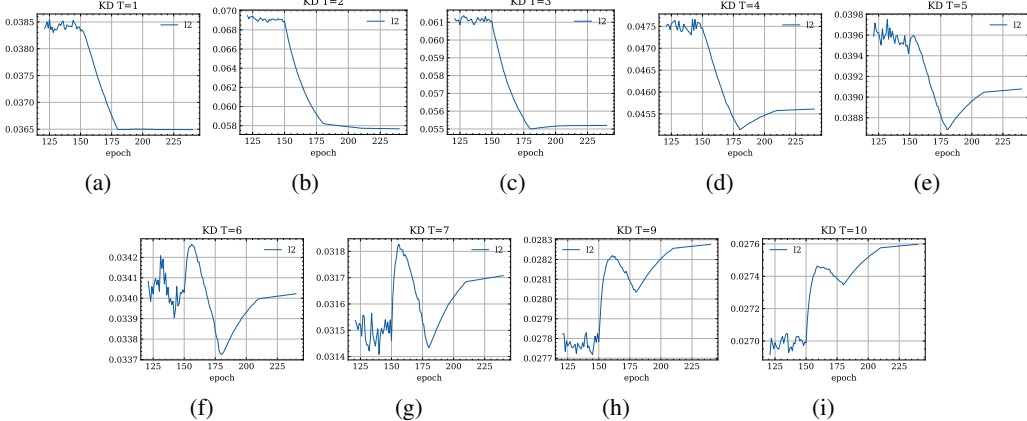

Figure 3: The L2 norm of the differences between weights of the classifier the student and teacher during training for different Temperatures (T).

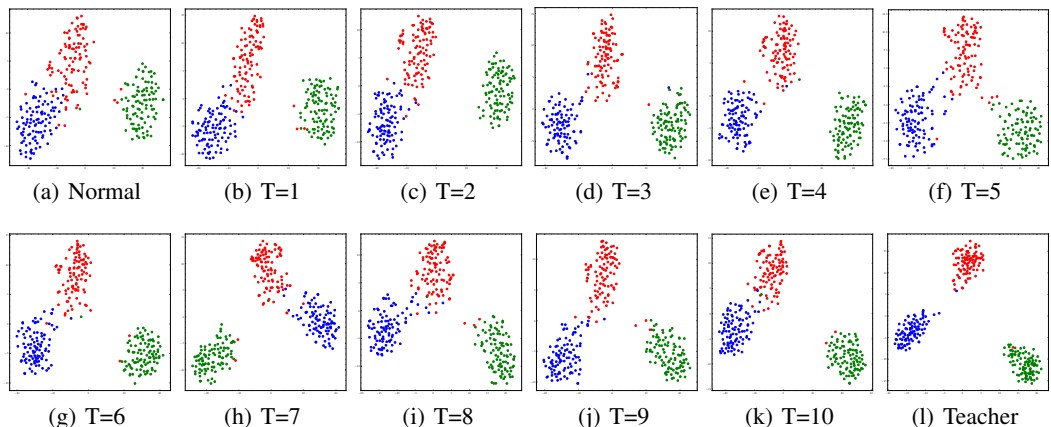

Figure 4: TSNE of the intermediate representations in the penultimate layer for different Temperatures.

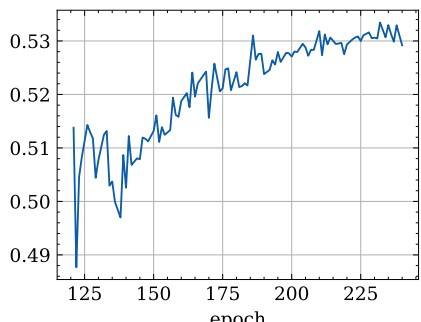

Figure 5: We see that the cosine similarity between representation vectors between a normally trained model and the teacher increases during training.

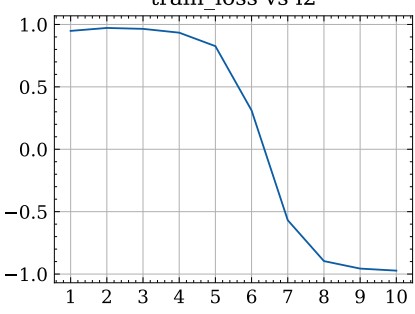

Figure 6: The plot of the pearson correlation coefficient between the train loss and the $L^2$ norm ($\mu$) for different temperatures

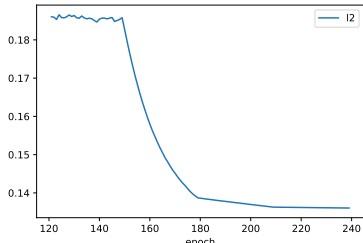

Figure 7: The plot of $\mu$ for the paper Knowledge Distillation from a Stronger Teacher (Huang et al., 2022).

## A.4 EXPERIMENTAL DETAILS

We adopt the CIFAR-100 dataset and two popular architecture ResNet 32x4 and ResNet 8x4, respectively as the teacher and student to conduct our experiments. All images are normalized by channel means and standard deviations. A horizontal flip is used for data augmentation. Each training image is padded by 4 pixels on each size and randomly cropped as a 32×32 sample. We use the SGD optimizer with 0.9 Nesterov momentum. We train for 240 epochs. The learning rate is set to 0.05 initially for the first 120 epochs, and after every 30 epochs afterwards, the learning rate is divided by 10. The weight decay is set to $5 \times 10^{-4}$.

