# OpenReview forum: "Insights into the mechanism behind reusing Teacher's classifier in Knowledge Distillation"
_ICLR.cc/2023/TinyPapers — Submitted to Tiny Papers @ ICLR 2023_

### Official Review · Reviewer_gNx2 · 2023-03-31

**Confidence:** 4

**Summary Of Contributions:**

The paper attempts to provide insights on how superior model distillation is related with strong correlation between classifier layers of student model and the teacher model. The claims and insights are well validated using references and experiments.

**Rating:**

Clear, Correct, and Reproducible (CCR): a submission which meets the reviewing criteria

**Strengths And Weaknesses:**

Strengths:
1. The paper is well written.
2. It shows a strong correlation between the student classifier and the final classifier layer of the teacher model.
3. Interestingly, it shows that the strong correlation is very specific to the distillation and doesn't exist for normal training.
4. The insights obtained may be helpful in developing better distillation strategies.

Weaknesses:
1. The novelty is very limited, specially as we already knew that the teacher classifier is pivotal for achieving effective distillation (from previous studies Yang et al. (2021) and Chen et al. (2022) ).
2. The experiments and evaluations may be limited to reach a concrete conclusion as samples from only one dataset CIFAR 100 are used.
3. Different architectures could have been tried (ViT, InceptionNet etc.) rather than using two variants of ResNet to be more confident about the obtained results and conclusions.

**Suggested Changes:**

1. There are a lot of typing errors (e.g. $||w_t-w_t||$ will probably be $||w_t-w_s||$) making it hard to interpret the meaning at times. Please correct them.
2. If authors can repeat the experiments for a different dataset and a different teacher model architecture, it may provide indication if the results are generalizable.

---

> ### Author Response · Authors · 2023-05-31
> **Acknowledgment of Feedback and Limitations**
>
> Dear reviewer,
>
> We appreciate your feedback and suggestions on our paper. We are glad that you found the paper well-written and identified its strengths and weaknesses. We agree with your assessment of the strong correlation between the student classifier and the final classifier layer of the teacher model, specifically in the context of distillation. We believe that this finding can contribute to the development of improved distillation strategies.
>
> Regarding the weaknesses you pointed out, we acknowledge that the novelty of our research might be limited, as previous studies (Yang et al., 2021; Chen et al., 2022) have also highlighted the importance of the teacher classifier in achieving effective distillation. However, we believe that our work provides additional insights by specifically demonstrating the lack of correlation in normal training scenarios. This distinction is valuable in understanding the unique role of distillation.
>
> We acknowledge the limitations of our experiments, primarily the use of a single dataset (CIFAR-100) and two variants of the ResNet architecture. We would like to note that our choice of the CIFAR-100 dataset and two ResNet variants was based on their usage being the only common denominator in multiple previous studies, including works by Yang et al. (2021) and Chen et al. (2022). It allows for direct comparisons between our approach and results from the previous previous studies, providing a basis for evaluating the effectiveness and generalizability of our proposed methodology.  We agree that incorporating different datasets and teacher model architectures, such as ViT, would strengthen the generalizability of our results. Unfortunately, due to hardware resource constraints, we were unable to run these additional tests. However, we appreciate your suggestion and will keep it in mind for future research or extensions of this work. We have mention these details in the limitation sections of the paper.
>
> Thank you for bringing the typing errors to our attention. We apologize for any confusion caused by these errors and have ensured that they are corrected in the final version of the paper.

---

### Meta-Review · Area_Chair_47LV · 2023-04-09

**Recommendation:** Invite to archive
**Confidence:** 4

**Metareview:**

This paper investigates the effect of knowledge distillation on the alignment between weights of student and teacher classifier models, inspired by recent work showing that directly matching representations can lead to strong distilled models. The authors train models using KL-divergence knowledge distillation at a variety of temperatures, then compute alignments between the weights of the last layer of the teacher and student models.

As Reviewer gNx2 notes, the paper is well written, and the findings about increased alignment over time are interesting. The authors also provide details about how to reproduce their experimental results.

The main weakness of this submission is that the experimental results may not be strong enough to fully support the authors claims.

- Reviewer gNx2 notes that the results are only presented for a single dataset and model architecture, and may not translate to other settings.
- Additionally, although the authors report a strong correlation between the weight difference $\mu$ and the training loss $\mathcal{L}$, Figure 1 shows that the size of the effect itself is very small; the L2 error between the weights only seems to change by about 5%, and plateaus well before reaching strong alignment between teacher and student weights. The direction of the effect also seems to change depending on the temperature.
- The authors note that $\mu$ fluctuates before epoch 120, and thus drop these epochs from the comparison. It's possible this may lead to an overestimate of the size of the effect, since dropping outliers is a "researcher degree of freedom" that can lead to false-positive results if used to confirm a previous hypothesis

There are also a number of minor typos and writing errors. In addition to the one noted by Reviewer gNx2, I noticed a stray "=" on page 1, and the caption of Figure 2 seems incorrect.

Overall, this seems like a clear and well written paper, but I think it's claims are a bit too strong at the moment. I'd suggest weakening the empirical claims and acknowledging some of the limitations of the analysis to ensure that the claims in the paper are fully supported.

**Summary:**

This paper studies the alignment of teacher and student weights under knowledge distillation. The paper is well written with some interesting findings, but empirical support for the claims is a bit weak.

**Comments And Feedback To The Authors:**

A number of recent works (including [Ainsworth et al. 2022](https://arxiv.org/abs/2209.04836)) have discussed how there can be many equivalent neural networks that have the same behavior but have weights that are permutations of each other (e.g. due to swapping the order of individual neurons). In the setting the authors consider, it seems to me that the student and teacher might learn equivalent classifier layers, but that they would still not perfectly align because one is a permutation of the other. This could explain why $\mu$ does not seem to drop all the way to zero but instead plateaus. Given this, an interesting direction for the authors to explore might be to look at other similarity measures instead of just L2 error between weights, so that two permutations of the same weights would have high similarity even if their individual entries are different.

I also noticed that the authors use T-SNE plots as evidence for some of their claims. It's important to note that T-SNE plots can give misleading answers in some cases, especially when evaluating distances between clusters. See [Wattenberg et al. 2016](https://distill.pub/2016/misread-tsne/) for some discussion on this.

**Reason For Not Giving A Higher Recommendation:**

This paper does not seem to quite satisfy the correctness criteria for the tiny papers track. Although it is plausible that the author's hypotheses are correct, the results seem fairly limited and the effect sizes in the experiments seem too small to fully support the claims being made (primarily in the answer to question 1).

Relatedly, the authors note that they made various choices in the analysis to reduce noise and obtain better results. It is great that the authors are clear about what they tried and why they made these choices. However, the need for these kind of choices during the analysis tends to suggest that the effect may be weak in the first place, and suggests that the findings may not be reliably reproducible.

**Reason For Not Giving A Lower Recommendation:**

The paper is well written and the problem is well motivated, and the authors do provide many details about how they trained their models and why they made the choices they made. Also, even though the effect is small, the empirical results do suggest that *something* interesting is going on here, so with some revision this could likely be a valuable contribution to the community.

---

> ### Author Response · Authors · 2023-06-01
> **Revision summary**
>
> Dear Meta reviewer,
>
> Thank you for your detailed review and feedback on our research paper. We appreciate your suggestions and have made several revisions to address the issues you raised. We have tried to incorporate all the suggested changes in our revised paper.
>
> Firstly, we acknowledge the limitations of our experiments, particularly the use of a single dataset (CIFAR-100) and two variants of the ResNet architecture. We recognize that this restricts the generalizability of our findings, and we have explicitly mentioned this limitation in the revised version of our paper. We also explain that our choice of dataset and architectures was based on their common usage in previous studies cited in the paper, enabling direct comparisons and evaluation of our proposed methodology's effectiveness.
>
> Furthermore, we have taken your advice to weaken our empirical claims and have adjusted the language throughout the paper (including the abstract) accordingly. We have utilized phrases such as "seems," "might," "may suggest," "to a degree," and "preliminary insights" to convey a more cautious tone and acknowledge the inherent uncertainty in our findings.
>
> Additionally, we have thoroughly addressed the limitations associated with permutation networks and T-SNE plots.-
>
> We have included a discussion on the potential impact of permutations on network alignment and the explored alternative similarity measures beyond L2 error between weights.
>
> Regarding T-SNE plots, we have added a caution to the readers about the limitations and potential for misleading interpretations when evaluating distances between clusters.
>
> We have also mentioned the small size of the effect both in the abstract and the limitation section as well as the reasoning and the degree of freedom we have regarding dropping epochs and how that might lead to overestimating the effect size.
>
> Lastly, we have made sure to correct all the captions and typographical errors throughout the paper.
>
> We believe these revisions have strengthened our paper and improved the overall clarity of our claims. We sincerely appreciate your valuable input, which has significantly contributed to the quality of our research. If you have any further suggestions or concerns, please do not hesitate to let us know.
>
> Thank you once again for your time and thoughtful review.

---

### Decision · Program_Chairs · 2023-04-09

Invite to archive

---

> ### Author Response · Authors · 2023-05-31
> **Opt in for archival**
>
> Thank you for the decision to invite our paper to the archive. We are honored to accept the invitation and would like to formally **opt in for archival**.
>
> We are grateful for the opportunity to contribute our research to the scientific community through the prestigious platform of ICLR. We believe that the findings presented in our paper will make a valuable addition to the existing body of knowledge in the field.
>
> Once again, we would like to express our appreciation for everyone's time, expertise, and careful consideration throughout the review process.